# Laboratory Intercomparison of Radiometers Used for Satellite Validation in the 400–900 nm Range

**Viktor Vabson [1,*], Joel Kuusk [1] , Ilmar Ansko [1], Riho Vendt [1], Krista Alikas [1], Kevin Ruddick [2], Ave Ansper [1] , Mariano Bresciani [3], Henning Burmester [4] , Maycira Costa [5], Davide D'Alimonte [6], Giorgio Dall'Olmo [7], Bahaiddin Damiri [8] , Tilman Dinter [9], Claudia Giardino [3] , Kersti Kangro [1], Martin Ligi [1], Birgot Paavel [10], Gavin Tilstone [7], Ronnie Van Dommelen [11], Sonja Wiegmann [9], Astrid Bracher [9] , Craig Donlon [12] and Tânia Casal [12]**

1    Tartu Observatory, University of Tartu, 61602 Tõravere, Estonia; joel.kuusk@ut.ee (J.K.);
     ilmar.ansko@ut.ee (I.A.); riho.vendt@ut.ee (R.V.); krista.alikas@ut.ee (K.A.); ave.ansper@ut.ee (A.A.);
     kersti.kangro@ut.ee (K.K.); martin.ligi@ut.ee (M.L.)
2    Royal Belgian Institute of Natural Sciences, 1000 Brussels, Belgium; kruddick@naturalsciences.be
3    National Research Council of Italy, 21020 Ispra, Italy; bresciani.m@irea.cnr.it (M.B.);
     giardino.c@irea.cnr.it (C.G.)
4    Helmholtz-Zentrum Geesthacht, Institute for Coastal Research, 21502 Geesthacht, Germany;
     henning.burmester@hzg.de
5    University of Victoria, Victoria, BC V8P 5C2, Canada; maycira@uvic.ca
6    Center for Marine and Environmental Research, University of Algarve, 8005-139 Faro, Portugal;
     davide.dalimonte@gmail.com
7    Plymouth Marine Laboratory, Plymouth PL1 3DH, UK; gdal@pml.ac.uk (G.D.); GHTI@pml.ac.uk (G.T.)
8    Cimel Electronique S.A.S, 75011 Paris, France; bahaiddin.damiri@univ-lille1.fr
9    Alfred Wegener Institute Helmholtz Centre for Polar and Marine Research, D-27570 Bremerhaven, Germany;
     Tilman.Dinter@awi.de (T.D.); Sonja.Wiegmann@awi.de (S.W.); Astrid.Bracher@awi.de (A.B.)
10   Estonian marine institute, University of Tartu, 12618 Tallinn, Estonia; birgot.paavel@ut.ee
11   Satlantic; Sea Bird Scientific, Bellevue, WA 98005, USA; rvandommelen@seabird.com
12   European Space Agency, 2201 AZ Noordwijk, The Netherlands; craig.donlon@esa.int (C.D.);
     tania.casal@esa.int (T.C.)
*    Correspondence: viktor.vabson@ut.ee; Tel.: +372-737-4552

**Abstract:**   An intercomparison of radiance and irradiance ocean color radiometers (The Second Laboratory Comparison Exercise—LCE-2) was organized within the frame of the European Space Agency funded project Fiducial Reference Measurements for Satellite Ocean Color (FRM4SOC) May 8–13, 2017 at Tartu Observatory, Estonia. LCE-2 consisted of three sub-tasks: 1) SI-traceable radiometric calibration of all the participating radiance and irradiance radiometers at the Tartu Observatory just before the comparisons; 2) Indoor intercomparison using stable radiance and irradiance sources in controlled environment; and 3) Outdoor intercomparison of natural radiation sources over terrestrial water surface. The aim of the experiment was to provide one link in the chain of traceability from field measurements of water reflectance to the uniform SI-traceable calibration, and after calibration to verify whether different instruments measuring the same object provide results consistent within the expected uncertainty limits. This paper describes the activities and results of the first two phases of LCE-2: the SI-traceable radiometric calibration and indoor intercomparison, the results of outdoor experiment are presented in a related paper of the same journal issue. The indoor experiment of the LCE-2 has proven that uniform calibration just before the use of radiometers is highly effective. Distinct radiometers from different manufacturers operated by different scientists can yield quite close radiance and irradiance results (standard deviation $s < 1\%$) under defined conditions. This holds when measuring stable lamp-based targets under stationary laboratory conditions with all the radiometers uniformly calibrated against the same standards just

prior to the experiment. In addition, some unification of measurement and data processing must be settled. Uncertaint  of radiance and irradiance measurement under these conditions largely consists of the sensor's calibration uncertainty and of the spread of results obtained by individual sensors measuring the same object.

**Keywords:** ocean color radiometers; radiometric calibration; indoor intercomparison measurement; agreement between sensors; measurement uncertainty

---

## 1. Introduction

Fiducial reference measurements of water reflectance are aimed to validate satellite data with requirement to provide metrological traceability to the SI units with related uncertainty estimates. These measurement uncertainties can arise from instrument specification, calibration and characterization, performance during field measurements due to various conditions of use and different targets, measurement protocol (including any corrections and assumptions), traceability of calibration sources to the primary SI standards. The present study focusses particularly on instrument performance and calibration, assessing whether different instruments freshly calibrated under uniform conditions and methods, but operated by different scientists, can produce consistent estimates within the estimated uncertainty limits when measuring stable radiance and irradiance targets in laboratory conditions. The results of this study are not limited to ocean color (OC) radiometry and are relevant to radiometers operating over 400–900 nm used in air for other applications, including field measurement of land surface reflectance.

An intercomparison of radiance and irradiance ocean color radiometers (The Second Laboratory Comparison Exercise—LCE-2) using stable incandescent lamp sources under controlled indoor conditions was conducted with the aim to provide one link in the chain of traceability from field measurements of water reflectance to the uniform SI-traceable calibration (Figure 1). Intercomparison of data produced by a number of independent radiometric sensors measuring the same object can assess the consistency of different results and their estimated uncertainties depending on the type of the sensor, the spectrum of measured radiation, the environmental conditions, and the particular method used for collecting and handling the measurement data [1,2]. This information can also serve for further elaboration of uncertainty estimation. Additionally, methodologies used by participants for the measurements and data handling can also be critically reviewed. For the LCE-2, a stepwise approach was chosen: first, the radiometric calibration of the sensors was conducted by the same calibration laboratory; second, indoor comparisons using various levels of radiance or irradiance measurements were performed in stable conditions similar to those during radiometric calibration; third, field measurements as described further in [3]. Traceability of the in-situ measurements to SI units is established by regular calibration of field radiometers. Thus, immediately before the comparison, Tartu Observatory (TO) performed consistent calibration of all participating radiometers in order to guarantee that differences in comparison results will not be primarily due to various calibration sources and/or calibration times. Radiometric calibration procedures including respective uncertainties have been, in general, well established over the last decades, tested by several intercomparisons [2,4], and also confirmed by the current experiment. Although during the measurement of stable radiation sources the observations in recorded time series are not always completely independent, their autocorrelation can be accounted for and analysis results (including determination of reference or consensus values) is rather straightforward. Small variability between individual sensors found during the current experiment confirms usefulness of the radiometric calibration performed at the same laboratory just before the comparisons.

According to [5] calibration is an operation that, under specified conditions, in a first step, establishes a relation between the quantity values with measurement uncertainties provided by

measurement standards and corresponding indications with associated measurement uncertainties and, in a second step, uses this information to establish a relation for obtaining a measurement result from an indication.

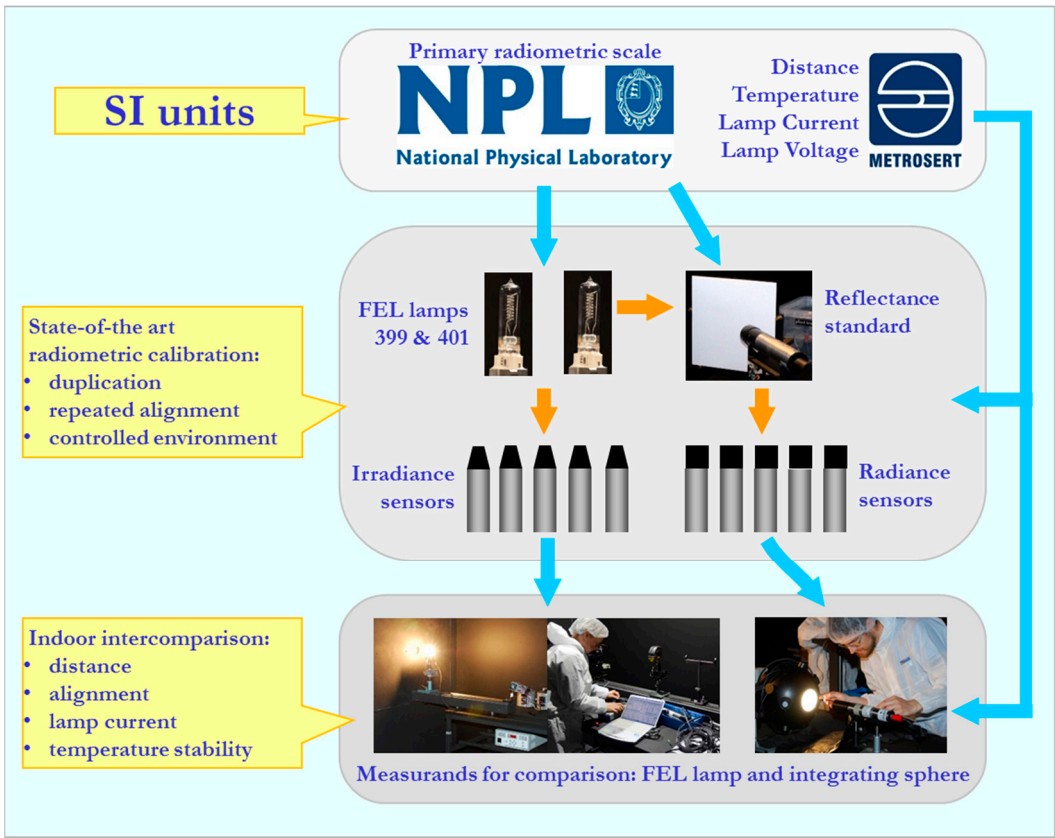

**Figure 1.** Traceability scheme of the LCE-2 for validation of indoor measurement uncertainties as specific to the present study.

For determination of spectral responsivity of a radiometer, it is usually calibrated against a known source placed at a specified distance from the entrance optics of the radiometer. Such a calibration procedure is well established and validated [6–12]. Unfortunately, specified conditions during the calibration may be quite different from varying conditions which may prevail during later use of the instrument. For radiometric sensors, there can be significant differences between calibration and later use in the field, in regards to the operating temperature, spectral variation of the target (giving different spectral stray light effects), angular variation of the light field (especially for irradiance sensors) and the intensity of the measured radiation. Each of these factors may interact with instrument imperfections to add further uncertainties when an instrument is used in the field and estimation of such uncertainties requires instrument characterization in addition to the well-established absolute radiometric calibration.

Instrument characterization, which can lead to corrections to reduce uncertainties, should include determination of thermal effects, nonlinearity, spectral stray light effects, wavelength calibration, angular response, and polarization effects. Procedures for determination of corrections, including measurement of all relevant influence quantities, are much less studied, and for some instruments often corrections might be not available. For applying corrections, individual testing of radiometers for each effect considered is indispensable. For most of the corrections, tests may be more time consuming than the radiometric calibration. Generally, the corrections should be applied both for calibration spectra and for field spectra calculated using the calibration coefficients, critically increasing the impact of data handling. Fortunately, these individual tests are carried out usually

only once in the lifetime of an instrument unit (i.e., a sensor from an instrument family/design with a unique serial number) while radiometric calibration has to be performed on a regular basis at least once a year. Methods for correcting temperature effects [2,4,13–17], spectral stray light effects [4,18–22], nonlinearity [4,15,17,23], and polarization effects [24] are the most studied. Nevertheless, difficulties may arise during the use of a calibrated instrument when some parameters influencing correction should be determined. Some radiometers do not have internal temperature sensors, and therefore, for these instruments the accuracy of temperature corrections is limited even when external temperature sensors are applied during the calibration and later use [2]. Nonlinearity effects present in calibration spectra can be determined rather satisfactorily, but it can be much more difficult to account for nonlinearity when instable natural radiation sources are measured. Effect due to response error of cosine collector [25,26] can be satisfactorily accounted for a well-known radiation source, but in the field conditions the angular distribution of radiation is often not known accurately enough for efficient correction of the cosine error.

This study aims to evaluate techniques and procedures needed for improvement of the traceability of the OC field measurements. In order to improve the consistency of measurements, in this work and in [3] unified and enhanced metrological specification of radiometers, additional individual testing procedures for relevant systematic effects of sensors, and procedures for unified data handling are discussed. Most of the instruments involved in LCE-2 were hyperspectral radiometers having hundreds or thousands of spectral bands and different spectral response functions. Even when the instruments are of the same type, they are not directly comparable to each other due to small manufacturing differences. For instance, each radiometer has individual spectral response function represented by a wavelength scale with different center wavelengths (CWL) of individual bands. Therefore, for comparing the data of all the instruments, a few Sentinel-3 Ocean and Land Color Instrument (OLCI) bands were selected and from the hyperspectral data the OLCI band values were retrieved. The intercomparison analysis was performed using the OLCI band values.

## 2. Material and Methods

### 2.1. Participants of the LCE-2

In total, 11 institutions were involved in the LCE-2, see Table 1. Altogether, 44 radiometric sensors from 5 different manufacturers were involved (Figure 2). The list of radiometers reflects the typical selection of instruments used for shipborne validation of satellite-derived water reflectance ('ocean color validation'). However, the number of each type of instrument is not necessarily representative of total validation data usage, since the SeaPRISM instrument is used by a multi-site network of autonomous systems [27], thus providing very significant quantities of validation data. As denoted by the combination "(2L, 1E)" in Table 1, most of the participating teams use an above-water field measurement protocol with three radiometers: two radiance sensors, for upwelling (water) and downwelling (sky) radiances, respectively, and an irradiance sensor, measuring downwelling irradiance. For the RAMSES and HyperOCR this is achieved by three separate devices, while the WISP-3 contains three spectroradiometers integrated into a single device, and the SR-3500 uses a single spectrometer equipped with interchangeable entrance optics for irradiance and radiance (and can, like all radiance sensors, be used sequentially to measure both upwelling and downwelling radiance). The SeaPRISM estimates irradiance (E) from direct sun radiance (L)—see [27]. In the scope of laboratory measurements, the multiple entrance optics of SR-3500 and WISP-3 were treated as separate radiometers. Technical parameters of the participating radiometers are given in Table 2.

Water reflectance can also be measured from underwater radiometers deployed either at fixed depths or during vertical profiles. Indeed, the RAMSES and HyperOCR designs (but not WISP-3, SR-3500, SeaPRISM) may also be used underwater. The present study is fully relevant for the calibration aspects of such radiometers in underwater applications, although extra characteristics, particularly immersion coefficients to transfer in-air calibrations to in-water [28] must also be studied.

**Table 1.** Institutes and instruments participating in the LCE-2 intercomparison.

| Participant | Country | L—Radiance; E—Irradiance Sensor |
|---|---|---|
| Tartu Observatory (pilot) | Estonia | RAMSES (2 L, 1 E) WISP-3 (2 L, 1 E) |
| Alfred Wegener Institute | Germany | RAMSES (2 L, 2 E) |
| Royal Belgian Institute of Natural Sciences | Belgium | RAMSES (7 L, 4 E) |
| National Research Council of Italy | Italy | SR-3500 (1 L, 1 E) WISP-3 (2 L, 1 E) |
| University of Algarve | Portugal | RAMSES (2 L, 1 E) |
| University of Victoria | Canada | OCR-3000 (OCR-3000 is the predecessor of HyperOCR) (2 L, 1 E) |
| Satlantic; Sea Bird Scientific | Canada | HyperOCR (2 L, 1 E) |
| Plymouth Marine Laboratory | UK | HyperOCR (2 L, 1 E) |
| Helmholtz-Zentrum Geesthacht | Germany | RAMSES (2 L, 1 E) |
| University of Tartu | Estonia | RAMSES (1 L, 1 E) |
| Cimel Electronique S.A.S | France | SeaPRISM (1 L) |

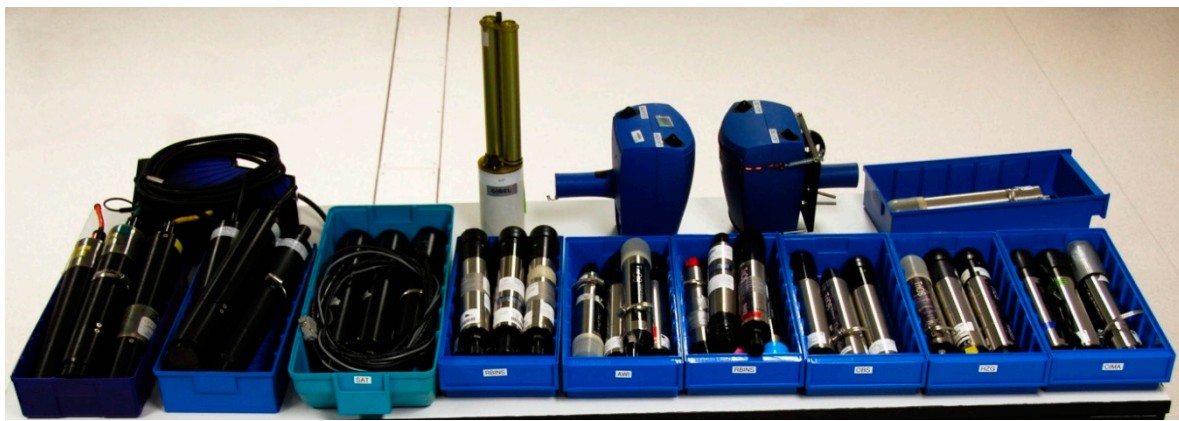

**Figure 2.** Instruments participating in the LCE-2 intercomparison.

**Table 2.** Technical parameters of the participating radiometers.

| Parameter | RAMSES | HyperOCR | WISP-3 | SR-3500 | SeaPRISM |
|---|---|---|---|---|---|
| Field of View (L/E) | 7°/cos | 6° (According to the manufacturer, the HyperOCR radiance sensors 444 and 445 have 6° FOV.) or 23°/cos | 3°/cos | 5°/cos | 1.2°/NA |
| Manual integration time | yes | yes | no | yes | no |
| Adaptive integration time | yes | yes | yes | yes | yes |
| Min. integration time, ms | 4 | 4 | 0.1 | 7.5 | NA |
| Max. integration time, ms | 4096 | 4096 | NA | 1000 | NA |
| Min. sampling interval, s | 5 | 5 | 10 | 2 | NA |
| Internal shutter | no | yes | no | yes | yes |
| Number of channels | 256 | 256 | 2048 | 1024 | 12 |
| Wavelength range, nm | 320...1050 | 320 … 1050 | 200 … 880 | 350 … 2500 | 400 … 1020 |
| Wavelength step, nm | 3.3 | 3.3 | 0.4 | 1.2/3.8/2.4 | NA |
| Spectral resolution, nm | 10 | 10 | 3 | 3/8/6 | 10 |

## 2.2. Calibration of Irradiance Sensors

A FEL type 1000 W quartz tungsten halogen spectral irradiance standard lamp was used for radiometric calibration of the radiometers [4]. The lamp was operated in constant current mode with a stabilized radiometric power supply Newport/Oriel 69935 ensuring proper polarity as marked on the lamp. A custom designed circuit was used for monitoring the lamp current through a 10 mΩ shunt resistor P310 and providing feedback to the power supply. Lamp current was stabilized to better than ±1 mA. The same feedback unit was used for logging the lamp current and voltage. Voltage was measured with a four-wire sensing method from the connector of the lamp socket. The power supply was turned on and slowly ramped up to the working current of the lamp. Calibration measurements were started after at least a 20-min warm-up time. The voltage across the lamp terminals was compared to the reference value measured during the last calibration of the lamp. A significant change in the operating voltage would have suggested that the lamp was no longer a reliable working standard

of spectral irradiance. In addition, the lamp's output was monitored by a two-channel optical sensor to detect possible short-term fluctuations. On completion of the calibration, the lamp current was slowly ramped down to avoid shocking the filament thermally.

The lamp and OC radiometer subject to calibration were mounted on an optical rail that passed through a bulkhead which separated the lamp and radiometer during calibration. A computer-controlled electronic shutter with a Ø60 mm aperture was attached to the bulkhead. The shutter was used for dark signal measurements during the calibration. Two additional baffles with Ø60 mm apertures were placed between the bulkhead and the radiometer at the distance of 50 mm and 100 mm from the bulkhead.

The OC radiometer subject to calibration and a filter radiometer next to it were mounted on a computer-controlled linear translation stage that allowed perpendicular movement with respect to the optical rail. Before calibration the positions of both radiometers were carefully adjusted and the translation stage positions saved in the controlling software. This allowed fast and accurate swapping of the radiometers after the lamp was turned on. In case of two groups of instruments (RAMSES and HyperOCR), several units having a common identical outside diameter in a group allowed a use of a V-block for fast mounting of the radiometers during the calibration. Before the lamp was turned on, the distance between the lamp and sensor was individually measured for each instrument, and a clamp was attached to fix the sensor at the appropriate position. During calibration, the radiometers of the same type were swapped without turning off the lamp. Placing the clamp against the end of the V-block ensured proper distance between the lamp and the radiometer during calibration.

The distance between the lamp and the radiometer was set with a custom designed measurement probe. One end of the probe was placed against the lamp socket reference surface and the other end of the probe had two lasers with beams intersecting at 120° angle. The intersection point defined, in a contactless way, the other end of the probe. Such design allowed distance measurement without touching the diffuser surface of the radiometer. The distance probe was calibrated by using a SI-traceable 500 mm micrometer standard. The uncertainty of distance determined with the probe was better than 0.2 mm.

The filter radiometer was used for monitoring possible long-term drifts in the optical output of the standard lamp. The filter radiometer was based on a three-element trap detector with Hamamatsu S1337-1010Q windowless Si photodiodes and temperature-controlled bandpass filters with peak transmittances at nominal wavelengths of 340 nm, 350 nm, 360 nm, 380 nm, 400 nm, 450 nm, 500 nm, 550 nm, 600 nm, 710 nm, 800 nm, 840 nm, 880 nm, 940 nm, and 980 nm. The photocurrent of the filter radiometer was amplified and digitized with a Bentham 487 current amplifier with integrating analog digital converter (ADC). A Newport 350B temperature controller was used for stabilizing the temperature of the bandpass filters. The filters were changed manually and it took about two minutes for the temperature of the filter to stabilize. Air temperature, relative humidity, and atmospheric pressure in the laboratory were recorded by a device located in the sensor compartment.

At least two different integration times were used for each radiometer (except in the case of the SeaPRISM and WISP-3 instruments for which the manufacturer-provided standard measurement programs were used). After a warm up time, at least 30 spectral measurements were collected measuring the radiation from the lamp. In the case of WISP-3 with internally selected integration time and averaging 10 spectra were collected. Next, the shutter in front of the lamp was closed and the same number of spectral measurements were collected, in order to estimate dark signal and ambient stray light in the laboratory. All measurements were repeated at least twice, including readjustment of the lamp and the sensor.

NPL provided two Gigahertz-Optik BN9101-2 FEL type irradiance calibration standard lamps with S/N 399 and 401 for the LCE-2 exercise in order to relate all measurements performed in the FRM4SOC campaigns to the common traceability source. The lamps were calibrated at the NPL and had not been used since the last calibration. Differences of the spectral flux of the two lamps in the range from 340 nm to 980 nm according to the aforementioned filter radiometer were within ±0.5%. The drift

of the irradiance values (at 500 nm) measured during the calibration campaign was ~0.1% which is close to the detection limit of the filter radiometer. In certificates issued for LCE-2 radiometers, arithmetic mean of the responsivities measured by the two lamps was used.

### 2.3. Calibration of Radiance Sensors

Radiance sensor calibration setup was based on the lamp/plaque method and utilized the components from the irradiance sensor calibration setup [4]. A Sphere Optics SG3151 (200 × 200) mm calibrated white reflectance standard was mounted on the linear translation stage next to the filter radiometer. Normal incidence for the illumination and 45° from normal for viewing were used. The panel was calibrated using the same illumination and viewing geometry at the NPL just before the LCE-2 exercise. A mirror in a special holder and an alignment laser were used for aligning the plaque and radiance sensor. As in the case of irradiance sensors, at least 30 calibration and background spectra were acquired using two different integration times (3 readings for SeaPRISM and 10 spectra for WISP-3). All measurements were repeated at least twice, including readjustment of the lamp, plaque, and radiometers.

### 2.4. Indoor Experiment of the LCE-2

The indoor experiment took place in the optical laboratory of TO within a few days after the radiometric calibration. The radiance and irradiance experiments were simultaneously set up and running during two days. Measurements were carried out by project participants under the supervision of TO's personnel.

#### 2.4.1. Irradiance Comparison Setup of the LCE-2

The irradiance setup can be seen in Figure 3. A FEL lamp was used as a stable irradiance source for indoor intercomparison. The power supply, current feedback unit, monitor detector, and distance measurement probe were the same as used during the radiometric calibration, but the FEL lamp and measurement distance were different. In order to change and align the radiometers without switching off the lamp, an additional alignment jig was placed between the shutter and the radiometer. When the shutter was closed, it was possible to change and realign the radiometer with respect to the jig. The alignment jig support was fixed to the optical rail during the whole intercomparison experiment and was used as a reference plane for distance measurement. During the intercomparison, the FEL source was switched off only once in the evening of May 9, the first day of the indoor exercise.

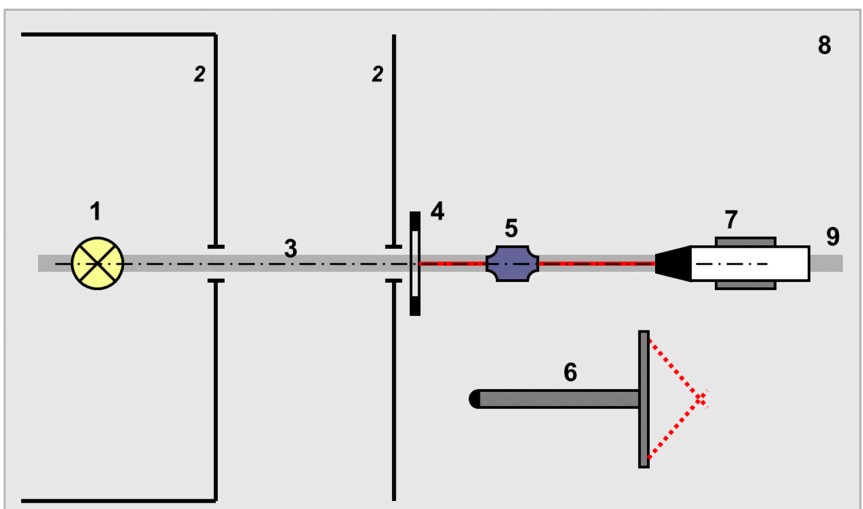

**Figure 3.** Indoor irradiance comparison. 1—FEL lamp; 2—baffles; 3—main optical axis; 4—alignment jig; 5—alignment laser; 6—distance probe; 7—radiometer on the support; 8—optical table; 9—optical rail.

Each participant measured the irradiance source using two different integration times (with corresponding dark series with the closed shutter) and one series with the instrument rotated by 90° around the optical axis. The measurement series were expected to contain at least 30 readings. As an exception, for the WISP-3 instruments two series (including re-alignment) of 10 readings were recorded.

### 2.4.2. Radiance Comparison Setup

The radiance setup for indoor intercomparison is depicted in Figure 4. A Bentham ULS-300 integrating sphere with internal illumination was used as the radiance source. ULS-300 is a Ø300 mm integrating sphere with Ø100 mm target port. According to the manufacturer, the uniformity of radiance over the output aperture is ±0.05% independent of the intensity setting. The sphere has a single 150 W quartz tungsten halogen light source (Osram Sylvania HLX 64640) and an eight-branch fiber bundle for transporting the light into the sphere. The sphere has a variable mechanical slit between the light source and the fiber bundle which allows changing the intensity of the light inside the sphere while maintaining the spectral composition of light which corresponds to correlated color temperature of (3100 ± 20) K. The lamp was powered by a Bentham 605 power supply at 6.3 A. A Gigahertz-Optik VL-3701-1 broadband illuminance sensor attached directly to the sphere was used as a monitor detector. The current of the monitor detector was recorded by an Agilent 3458A multimeter and the lamp voltage was measured by a Fluke 45 multimeter. Each participant measured the sphere source at two radiance levels and two distances from the sphere. The current reading of the monitor detector was used for setting the same sphere radiance levels for all the participants. For low radiance measurements, 1 µA monitor current was used corresponding roughly to the typical water radiance at 490 nm during field measurements, whereas 10 µA monitor current was used to simulate typical sky radiance. Obviously, the spectral composition of the incandescent sphere source did not match the field spectra, but was rather similar to the emission of the FEL-type radiometric calibration standard. In addition to sphere radiance, dark measurements were recorded by placing a black screen between the sphere and the radiometer. The sphere radiance was measured at two distances, typically 17 cm and 22 cm from the sphere port. Although the radiance measurement should not depend on measurement distance as long as the sphere port overfills the field-of-view (FOV) of the radiometer, the results measured at two distances were used to estimate the uncertainty component caused by back-reflection from the radiometer into the sphere.

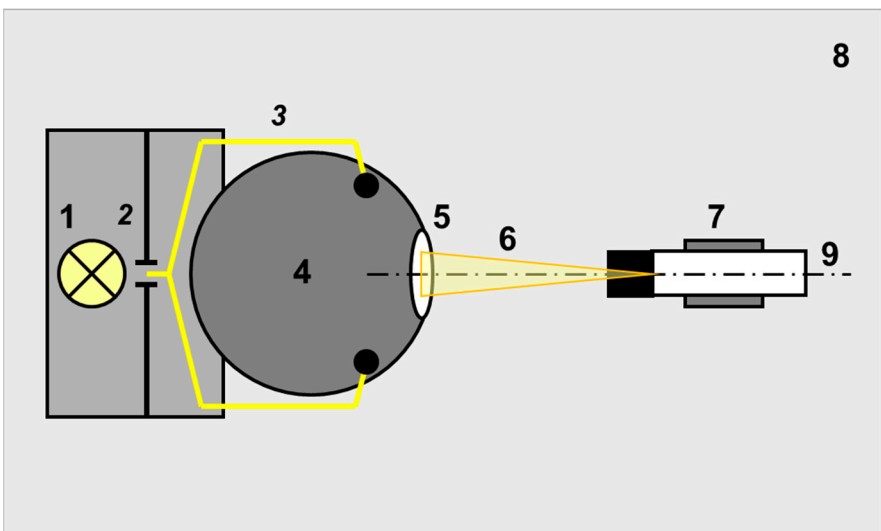

**Figure 4.** Indoor radiance comparison. 1—quartz tungsten halogen lamp; 2—variable slit; 3—optical fiber; 4—integrating sphere; 5—output port; 6—FOV of the radiometer; 7—radiometer on the support; 8—optical table; 9—main optical axis.

## 3. Results

### 3.1. Data Handling

The measurement results, including measurement uncertainty and information about measurement parameters, were reported back to the pilot laboratory in the form of spreadsheet files by most of the participants (for 33 out of the 44 sensors involved). For the rest, the pilot carried out the data analysis based on the raw instrument data. In the case of discrepancies, the pilot repeated the calculations on raw user data applying unified data handling described in the next chapter. Nevertheless, due to differences in hardware and software of the participating radiometers, fully unified data handling was not possible.

The participants were encouraged to perform the data processing for their radiometers and report back the radiance/irradiance values with uncertainty estimates. However, in a few cases, TO accomplished/repeated the calculations for some participants as well. Data processing for the RAMSES, HyperOCR, and WISP-3 instruments was fully automated at TO by purpose-designed computer software. The source code of the software is freely available for the participants.

Data processing of each instrument was performed independently from the others and included the following steps:

- separation of the raw datafiles based on the scene (e.g. low/high radiance, distance), integration time, shutter measurements;
- pairing the raw data with corresponding shutter measurement;
- dark signal subtraction;
- linearity correction whenever applicable;
- division by radiometric responsivity;
- recalculation for the OLCI spectral bands;
- averaging;
- evaluation of the uncertainty.

### 3.2. Device-Specific Issues

TriOS RAMSES series instruments include both the radiance (ARC) and irradiance (ACC) sensors. The raw spectra are stored in American Standard Code for Information Interchange (ASCII) and/or Microsoft ACCESS database files. Data processing for these radiometers is fully unified based on the measured data (2-byte integer numbers) and calibration files provided by the manufacturer and TO. The detailed procedure to derive the calibrated results is described in [4]. RAMSES instruments are equipped with black-painted pixels on the photodiode array used to compensate for the dark signal and electronical drifts. The background spectrum (with the external shutter closed) was subtracted as well. For subtraction, only the spectra with matching integration times were used. Before division by the responsivity coefficients, linearity correction was applied, see Section 4.1.4.

Satlantic HyperOCR/OCR3000 series instruments include also both the radiance and irradiance sensors with similar data processing chain. The raw spectra stored in binary files were converted to ASCII by participants using the proprietary manufacturer's software. Data processing for the HyperOCR was based on the calibration file provided by TO and is similar to the RAMSES procedure. The HyperOCR radiometers are equipped with an internal mechanical shutter, deployed automatically after every fifth target spectrum. The shutter measurements were detected in the datafiles and the closest shutter measurement was subtracted from each raw spectrum before the next steps.

Water Insight WISP-3 contains a three-channel Ocean Optics JAZ module spectrometer and computer. Two of the input channels are connected to the radiance inputs while the third is attached to the irradiance adaptor. Users can start the acquisition of the spectra by pressing a button, the internal computer is setting the measurement sequence, determining the integration times, and storing the data. All three channels are acquired simultaneously and the data are stored into

a single ASCII file. The spectrometers have painted detector array pixels like the RAMSES radiometers. The internal dark signal is subtracted automatically and resulting data are stored in the form of floating point numbers. The only operation needed was the division by the responsivity coefficients determined by TO using the same manual measurement sequence. The linearity correction described in Section 4.1.4 was not used.

Spectral Evolution SR-3500 spectrometer is equipped with an optical fiber input and interchangeable radiance and irradiance fore-optics. Thus, the data processing for the radiance and irradiance measurements are identical. The spectral output is stored in the ASCII files and can contain both the raw and radiometrically calibrated results based on the internal calibration coefficients. The dark signal is subtracted internally using an integrated mechanical shutter. Each target measurement is automatically followed by a dedicated dark measurement. During the radiometric calibration at TO, calibration factors to the existing coefficients were derived. The calibrated data in the files was multiplied by these factors, and finally, the linearity correction as described in Section 4.1.4 was used.

CIMEL SeaPRISM binary output was converted by the owner of the radiometer and was returned to the pilot in the form of ASCII files. Based on these data, TO derived the radiometric calibration coefficients. Neither linearity correction scheme nor re-calculation for the OLCI spectral bands was used for the SeaPRISM at this stage.

### 3.3. Calculation of Sentinel-3/OLCI Band Values

As the final step of data processing, the radiance and irradiance values were re-calculated for the OLCI spectral bands for each radiometer except for the multispectral SeaPRISM, in which case the initial band values were used. Based on the given CWL of the spectroradiometer $\lambda_n$, and the OLCI band definition $O_i(\lambda)$ [29], the weight factors were found for each pixel

$$C_i(n) = O_i(\lambda_n), K_i(n) = \frac{C_i(n)}{\sum_k C_i(k)}, \tag{1}$$

where $n$ is the pixel number with CWL of $\lambda_n$, $O_i(\lambda_n)$ is the responsivity of the corresponding $i$th OLCI band interpolated to $\lambda_n$, and $K(n)$ is the normalized weight coefficient for $n$'th pixel. Finally, the radiance/irradiance value $I_i$ for the corresponding OLCI band was calculated as

$$I_i = \sum_n I(n) \cdot K_i(n), \tag{2}$$

where $I(n)$ denotes the measured radiance/irradiance at the $n$'th pixel.

### 3.4. Consensus and Reference Values Used for the Analysis

Consensus values were calculated as median [30] of all presented comparison values. Reference values were applicable only for the indoor irradiance measurements (Figure 8), when the measurand used for this exercise was during comparison measured also with the precision filter radiometer serving as a reference.

### 3.5. Results of Indoor Experiment

The comparison results are presented as deviation from the consensus value. Despite the different sensor types, as the radiation sources used for indoor comparison were spectrally very similar to calibration sources, agreement between sensors was satisfactory for radiance and for irradiance sensors (Figures 5–8) with no outliers present. In these figures, blue dashed lines show the expanded uncertainty covering 95% of all data points on the right graphs. Solid lines represent RAMSES sensors, dashed lines—HyperOCR sensors, double line—SR-3500, and dotted lines—WISP-3 sensors.

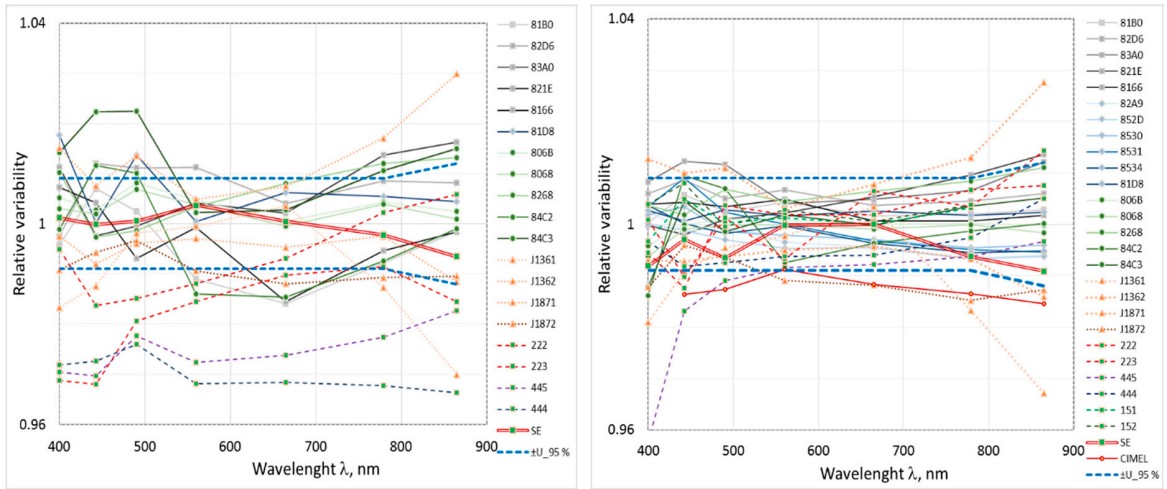

**Figure 5.** Low intensity radiance; agreement just after receiving data from participants (left), and after reviewing data by pilot, corrections submitted by participants and/or unified data handling by pilot (right). Blue dashed lines—expanded uncertainty covering 95% of all data points on the right graph. Solid lines—RAMSES sensors; dashed lines—HyperOCR sensors; double line—SR-3500; dotted lines —WISP-3 sensors.

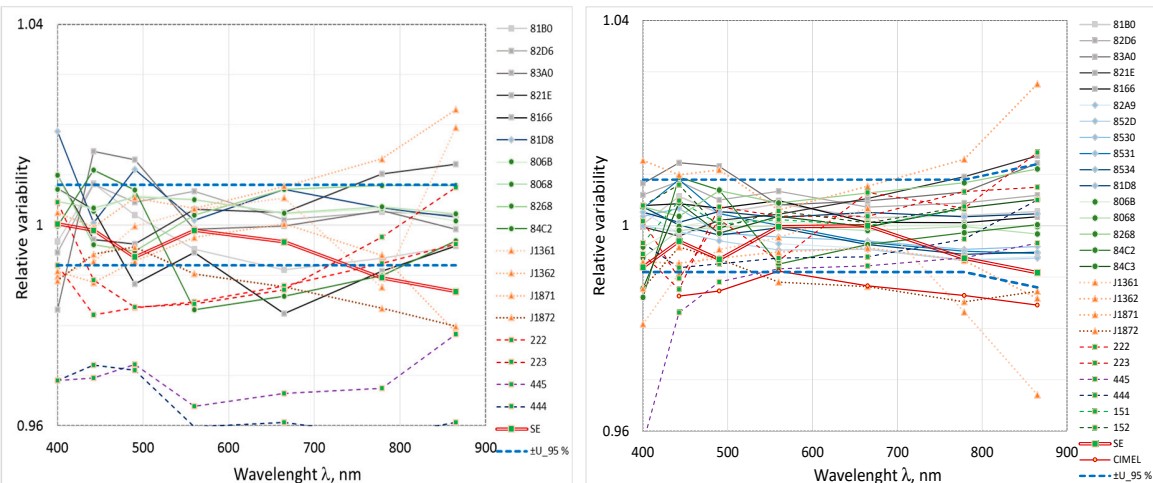

**Figure 6.** High intensity radiance; agreement just after receiving data from participants (left), and after reviewing data by pilot, corrections submitted by participants and/or unified data handling by pilot (right).

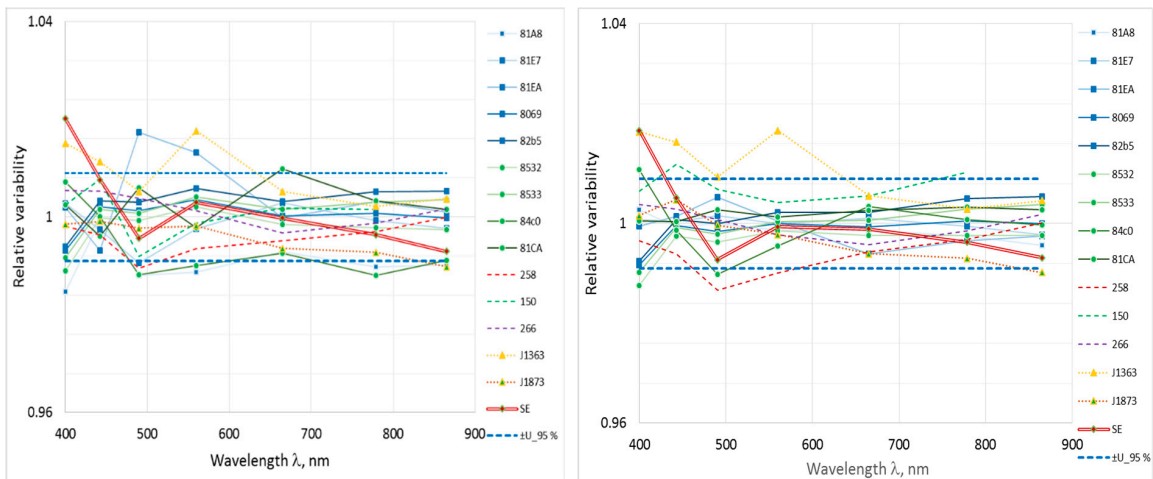

**Figure 7.** Irradiance sensors; agreement just after receiving data from participants (left), and after reviewing data by pilot, corrections submitted by participants and/or unified data handling by pilot (right).

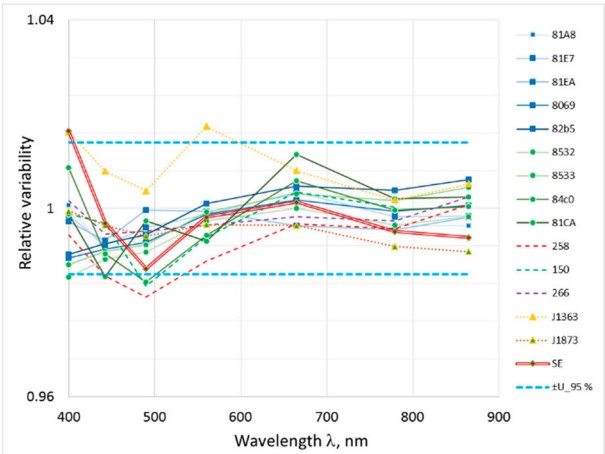

**Figure 8.** Irradiance sensors; agreement with reference values of the filter radiometer. Blue dashed lines—expanded uncertainty covering 95% of all data points. Uncertainty of radiometric calibration is included.

Larger variability of the results initially reported by participants was caused by applying out-of-date calibration coefficients, by diversely applying or not applying the non-linearity correction (Section 4.1.4) or calculating the OLCI band values differently. For unified data handling carried out by the pilot and described in 3.1 to 3.3, the calibration results obtained during LCE-2 were used, non-linearity correction was applied, OLCI band values were calculated by using individual weights as determined from the wavelength scale of each radiometer. After unified data handling, agreement between comparison results was significantly improved for the radiance sensors (Figures 5 and 6). There was almost no improvement in the case of the irradiance sensors in Figure 7.

## 4. Measurement Uncertainty

The uncertainty analysis has been carried out according to the ISO Guide to the Expression of Uncertainty in Measurement [31]. The evaluation is based on the measurement model, which describes the output quantity $y$ as a function $f$ of input quantities $x_i$: $y = f(x_1, x_2, x_3 \dots )$. Standard uncertainty is evaluated separately for each input quantity. There are two types of standard uncertainties: Type A is of statistical origin; Type B is determined by any other means. Both types of uncertainties are indicated as standard deviation and denoted by $s$ and $u$ respectively. The uncertainty component arising from averaging a large number of repeatedly measured spectra of radiation sources by array spectrometers

is considered as Type A. Contributions from calibration certificates (lamp, current shunt, multimeter, diffuse reflectance panel, etc.), but also from instability and spatial non-uniformity of the radiation sources are considered of Type B. For all input quantities relative standard uncertainties are estimated. The relative combined standard uncertainty of output quantity is calculated by combining relative standard uncertainty of each input estimate by using Equation (12) in [31]. Uncertainty of the final result is given as relative expanded uncertainty with a coverage factor $k = 2$.

### 4.1. Effects Causing Variability of the Results

### 4.1.1. State of Radiometric Calibration

Analysis of the LCE-2 calibration results, comparing them with former calibrations, including the factory calibrations, and also with calibrations carried out on the same set of radiometers by TO one year later (before the FRM4SOC FICE-AAOT intercomparison) demonstrates the importance of radiometric calibration for SI traceable results and reveals interesting information about instability of the sensors. Some uncertainty contributions characteristic to calibration can also be estimated.

The variability of calibration coefficients of radiance and irradiance sensors due to adjustment of the lamps, plaques, and sensors, and due to short-term instability of the lamps and sensors is depicted in Figure 9. All the radiometers were calibrated before LCE-2 using the same pair of lamps (Sections 2.2 and 2.3). Two sets of calibration coefficients were obtained for each sensor and the difference between the lamps was presented as the ratio of these coefficients. The curves in Figure 9 are calculated as standard deviations from the ratios of a whole set of calibration coefficients determined by using the two standard lamps. The systematic difference between lamps (due to small difference in traceability to SI) is neglected and only the other uncertainty components related to individual setting up and measurement of radiometers are accounted for by using standard deviation. Data in Figure 9 include calibration of more than 25 sensors for LCE-2 intercomparison and also for FICE-AAOT intercomparison one year later when a different pair of lamps was used. Remarkable in Figure 9 is the rapid increase of variability between sensors in the UV region.

Figure 10 shows average long-term variability of calibration coefficients of TriOS RAMSES and Satlantic HyperOCR radiance and irradiance sensors. All the radiometers had previous radiometric calibration certificates of various origin and age. The curves in Figure 10 are calculated similarly to Figure 9 as standard deviations of the ratios of previous and the last calibration coefficients. It has to be noted, however, that in this case standard deviation is characterizing dispersion between previous calibrations as these were performed by using various standards and conditions. Many of the RAMSES and HyperOCR radiometers that participated in LCE-2 also took part in the FICE-AAOT field intercomparison experiment one year later. Those sensors were radiometrically calibrated again at TO in June 2018 before the beginning of the field campaign. This gave a good opportunity to estimate the long-term stability of the sensors while minimising other possible factors influencing the calibration result. The sensors were calibrated in the same laboratory by the same operator in similar environmental conditions using the same calibration setup and methodology. Only the calibration standard lamps were exchanged since LCE-2. Nevertheless, the $L\_1$ yr and $E\_1$ yr curves in Figure 10 obtained as standard deviations of the ratios of the calibrations coefficients one year apart exclude the systematic differences between lamps. The two calibrations done in the same lab one year apart showed that over 80% of the sensors have changed less than ±1%. Thus, the inherent long-term stability of the sensors is likely better than 5% to 10% revealed from the previous calibration history, where the differences were likely caused by other factors such as different calibration standards, environmental conditions, calibration setups and methodologies, etc. However, rapid changes in the responsivity of some TriOS RAMSES irradiance sensors may cause even larger deviations which cannot be explained by other factors than the instability of the sensor itself. No quick changes were observed for the RAMSES radiance sensors, however, even after omitting outliers from the stability data of irradiance sensors, the stability of RAMSES radiance sensors is still better.

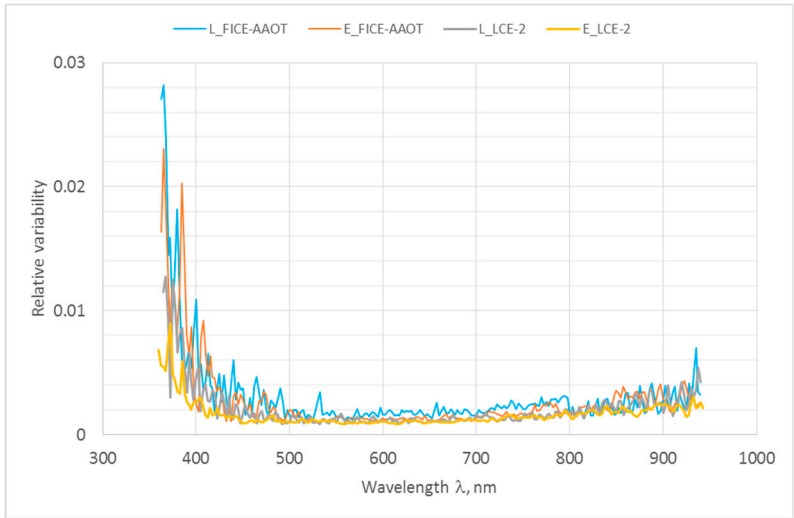

**Figure 9.** Relative variability of calibration coefficients of radiance (*L*) and irradiance (*E*) sensors with two different lamps used for calibration before LCE-2 and a year later before FICE-AAOT.

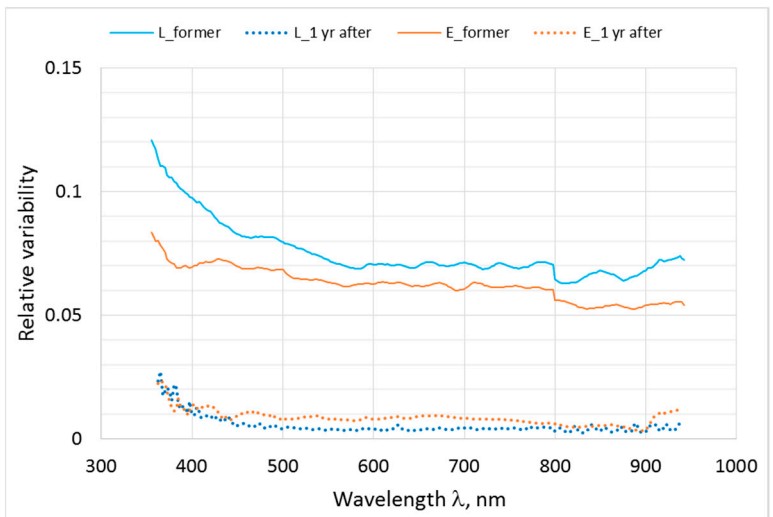

**Figure 10.** Relative variability of calibration coefficients of radiance (*L*) and irradiance (*E*) sensors: former—difference of previous known calibrations and results of LCE-2 calibration; 1 yr after— changes during one year after LCE-2 calibrations, some extra-large changes excluded.

### 4.1.2. Abrupt Changes of Responsivity

Factors causing the variability in the responsivity of radiometers were listed in [4]. During the calibration, the uncertainty of the radiation source is the dominant component in the uncertainty budget, assuming that usually the ambient temperature will be within ±1 °C. Based on the experience from LCE-2 and the following FICE activities, differences smaller than ±2% in the wavelength range of (350...900) nm can be observed between different sources used for calibration. Nevertheless, in some cases sharp changes in the responsivity of radiometers were detected, substantially exceeding all possible effects which can cause variability during calibration like the radiation source, alignment of instruments, contamination of fore-optics, temperature effects, etc. Relative change of the spectral irradiance responsivity of the TriOS RAMSES SAM_8329 10 times calibrated during eight years period is depicted in Figure 11. Each calibration in 2016–2018 consists of three repetitions conducted in a short time.

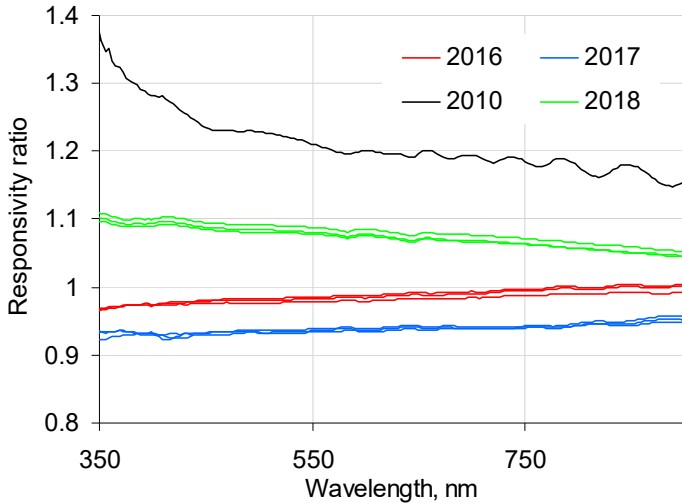

**Figure 11.** Relative change of responsivity of the SAM 8329. Year of the radiometric calibration is shown with color: 2010 black, 2016 red, 2017, blue, 2018 green.

### 4.1.3. Temperature Effects

Individual variation of the calibration coefficients as a function of temperature for each radiometer was not determined because of limited time schedule. Temperature effects for the TriOS RAMSES radiometers were evaluated based on [13] instead, see Figure 12.

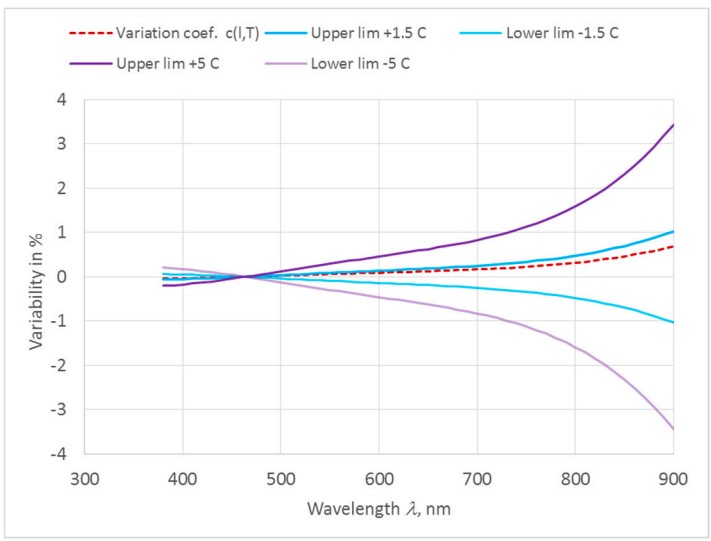

**Figure 12.** Relative variability of calibration coefficients due to temperature deviations from the reference temperature 21.5 °C.

### 4.1.4. Nonlinearity Due to the Integration Time

Maximum relative nonlinearity effect due to integration times determined from calibration spectra of TriOS RAMSES radiometers remained in the range of (1.5...3.5)% (Figure 13). Variability between the instruments due to this effect, if not corrected, will mostly be in the range of ±1%. The effect can be corrected down to 0.1% for certain types of radiometers by using the special formula, provided that there are at least two spectra with different integration times available for the same source. Derivation of the correction formula is based on the following assumptions: i) the nonlinearity effect is zero for the dark signal; ii) the effect is proportional with the recorded signal; iii) the effect is wavelength dependent; and iv) the corrected signal does not depend on the initial spectra used for estimation,

i.e., it should be of the same size for all possible combinations of initial spectra. Linearity corrected raw spectrum $S_{1,2}(\lambda)$ is calculated as

$$S_{1,2}(\lambda) = \left[1 - \left(\frac{S_2(\lambda)}{S_1(\lambda)} - 1\right)\left(\frac{1}{t_2/t_1 - 1}\right)\right]S_1(\lambda). \tag{3}$$

Here $S_1(\lambda)$ and $S_2(\lambda)$ are the initial spectra measured with integration times $t_1$ and $t_2$. Minimal ratio is usually $t_2/t_1 = 2$, but may be also 4, 8, 16, etc. For large ratios $t_2/t_1 > 8$ the spectrum $S_1(\lambda)$ is close to corrected spectrum $S_{1,2}(\lambda)$ and application of nonlinearity correction is not needed. Uncertainty of corrected spectrum is predominantly determined with the uncertainty of initial spectrum measured with smaller integration time. Therefore, the smallest uncertainty of the corrected spectrum will be obtained if the initial spectra with the largest and with the second-largest non-saturating integration times are used for estimation.

The formula has been found to perform quite effectively for TriOS RAMSES and Satlantic HyperOCR radiometers in the range of (400 . . . 800) nm. This nonlinearity correction method is not recommended for outdoor measurements, as due to temporal variability of the natural radiation consecutive measurements with different integration times may lead to uncertainty of the corrected spectrum much larger than acceptable 0.2%.

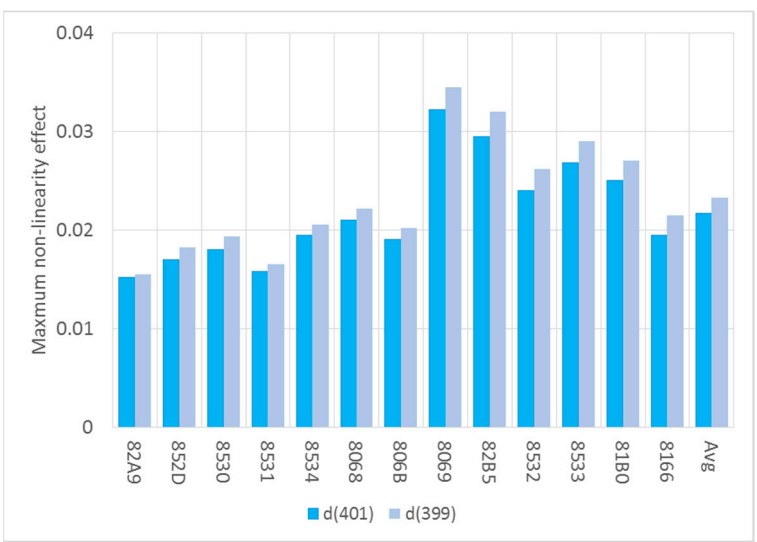

**Figure 13.** Maximum relative nonlinearity effect determined for 14 RAMSES sensors (both radiance and irradiance) from calibration spectra with FEL lamps 399 and 401.

From the analysis of the calibration data by using the two-spectra formula (3), it became evident that non-linearity errors scaled to full-range value of different radiance sensors behave in similar way. This behavior serves as a basis for derivation of nonlinearity correction applicable to a particular single spectrum that can also be used for the outdoor measurements, see Figure 14.

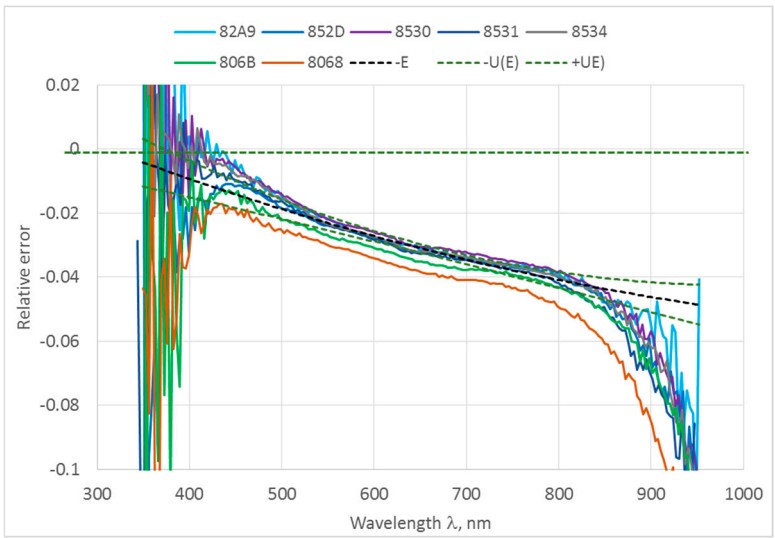

**Figure 14.** Non-linearity errors of different radiance sensors scaled to full-range value. Dashed lines are fitted model with uncertainty.

Relative nonlinearity correction for the full range signal $\delta x_{\max}(\lambda)$ is

$$\delta x_{\max}(\lambda) \; = \; -5.1 \cdot 10^{-8} \lambda^2 + 0.00014 \cdot \lambda - 0.0355. \tag{4}$$

Relative nonlinearity correction $\delta x(\lambda)$ for the signal $x(\lambda)$ is

$$\delta x(\lambda) \; = \; \frac{x}{x_{\max}} \delta x_{\max}(\lambda). \tag{5}$$

Corrected signal $x_{\mathrm{cor}}(\lambda)$ can be expressed as

$$x_{\mathrm{cor}}(\lambda) \; = \; x(\lambda)\Big[1 + \frac{x}{x_{\max}} \delta x_{\max}(\lambda)\Big]. \tag{6}$$

The formula has been thoroughly tested on the TriOS RAMSES calibration data, and is effective in the range of (400 . . . 800) nm of correcting nonlinearity mostly better than to 0.2%. The model can be fitted to all the studied RAMSES instruments by adjusting only the constant term.

### 4.1.5. Spectral Stray Light Effects

For many measurements, spectral stray light can lead to significant distortion of the measured signal and become a significant source of uncertainty [18,28]. Iterative technique [18,32] can be used for the simultaneous correction of bandpass and stray-light effects. When the full spectral stray light matrix (SLM) of a spectrometer is known, the stray light contribution can be removed from the measured signal and the original source spectrum restored. The stray light correction for a remote sensing reflectance measurement made by a common three-radiometer above-water system means that altogether six raw spectra have to be corrected—two for each radiometer, because stray light correction needs to be applied also for the standard source spectrum during the radiometric calibration.

The SLM was known for some radiometers from previous characterization such as for RAMSES sensors of TO, and for HyperOCR sensors of Plymouth Marine Laboratory (PML). Figure 15 presents the impact of stray light correction, evaluated for indoor measurements. The indoor radiance and irradiance sources were spectrally similar to the calibration sources; therefore, the stray light correction has relatively small impact. WISP-3, SR-3500, and SeaPRISM have different optical design, thus, their spectral stray light properties can have different nature compared to the data presented in Figure 15.

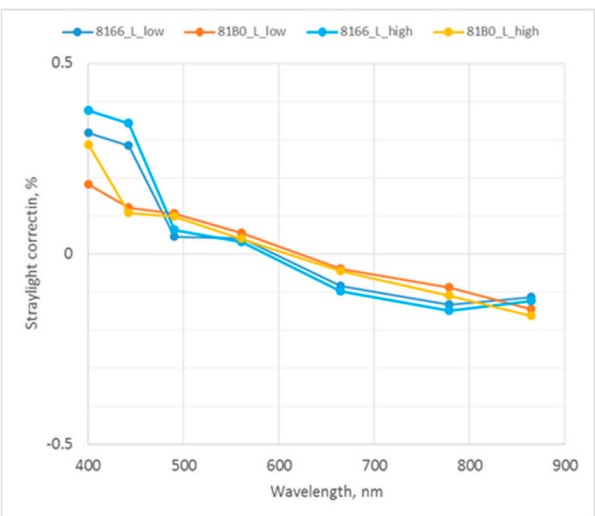

**Figure 15.** Stray light effects for indoor radiance measurements. Two RAMSES radiance sensors at high and low sphere radiance.

## 4.2. Uncertainty Budgets for Indoor Comparisons

Uncertainty analysis is made for the laboratory irradiance and radiance measurements. Uncertainty estimates for irradiance sensors measuring an FEL source at approximately 1 m distance are given in Table 3, and for radiance sensors measuring integrating sphere, in Table 4. All the uncertainty estimations of TriOS RAMSES sensors besides experimental data are based on information from [2,24,28,33–36]. For the other radiometer models that took part in the intercomparison, very little publicly available information can be found regarding various instrument characteristics that influence the measurement results [37]. In addition, the RAMSES was the only sensor model that was represented in sufficiently large number for statistical analysis.

The uncertainty is calculated from the contributions originating from the spectral responsivity of the radiometer, including data from the calibration certificate, from interpolation of the spectral responsivity values to the designated wavelengths and/or spectral bands, from instability of the array spectroradiometer, from contribution to the spectral irradiance and/or radiance due to setting and measurement of lamp current, from measurement of the distance between the lamp and input aperture of the radiometer, from the spatial uniformity of the irradiance at 1 m distance, and from reproducibility of the alignment. For the radiometer, uncertainty contributions arising from the non-linearity, temperature effects, spectral stray light, and from dark measurements, as well as from repeatability and reproducibility of averaged signal are included.

The radiometric calibration uncertainty in the following tables is not added to the combined uncertainty because all participating radiometers were calibrated using common standards shortly before the intercomparison and the contribution from calibration when using unified data handling does not affect the relative differences of participants to each other. The following uncertainty budgets describe variability between individual sensors, while uncertainty of radiometric calibration and contributions of systematic effects which influence the instruments in a similar way are not accounted for. Nevertheless, certainly they are relevant for traceability to SI units.

**Table 3.** Relative uncertainty budget for the irradiance (in percent), based on spread of individual sensors measuring the same lamp during indoor comparison. Data highlighted in green are not used for combined and expanded uncertainties.

|                      | 400 nm | 442.5 nm | 490 nm | 560 nm | 665 nm | 778.8 nm | 865 nm |
|----------------------|--------|----------|--------|--------|--------|----------|--------|
| Certificate          | 0.88   | 0.68     | 0.65   | 0.62   | 0.59   | 0.62     | 0.56   |
| Interpolation        | 0.5    | 0.2      | 0.3    | 0.2    | 0.2    | 0.1      | 0.1    |
| Instability (sensor) | 0.05   | 0.03     | 0.04   | 0.03   | 0.04   | 0.03     | 0.02   |
| Alignment            | 0.1    | 0.1      | 0.1    | 0.1    | 0.1    | 0.1      | 0.1    |
| Nonlinearity         | 0.2    | 0.15     | 0.15   | 0.15   | 0.15   | 0.15     | 0.2    |
| Stray light (sensor) | 0.2    | 0.2      | 0.2    | 0.2    | 0.2    | 0.2      | 0.2    |
| Temperature          | 0.02   | 0.01     | 0.01   | 0.03   | 0.09   | 0.2      | 0.38   |
| Instability (source) | 0.14   | 0.14     | 0.12   | 0.11   | 0.1    | 0.09     | 0.08   |
| Uniformity           | 0.1    | 0.1      | 0.1    | 0.1    | 0.1    | 0.1      | 0.1    |
| Stray light (source) | 0.1    | 0.1      | 0.1    | 0.1    | 0.1    | 0.1      | 0.1    |
| Signal, type A       | 0.11   | 0.04     | 0.02   | 0.02   | 0.01   | 0.02     | 0.04   |
| Combined ($k = 1$)   | 0.63   | 0.39     | 0.45   | 0.38   | 0.39   | 0.39     | 0.52   |
| Expanded ($k = 2$)   | 1.3    | 0.8      | 0.9    | 0.8    | 0.8    | 0.8      | 1.0    |

**Table 4.** Relative uncertainty budget for the radiance (in percent) based on spread of individual sensors measuring the same integrating sphere during indoor comparison. Data highlighted in green are not used for combined and expanded uncertainties.

|                      | 400 nm | 442.5 nm | 490 nm | 560 nm | 665 nm | 778.8 nm | 865 nm |
|----------------------|--------|----------|--------|--------|--------|----------|--------|
| Certificate          | 1.2    | 0.78     | 0.76   | 0.73   | 0.71   | 0.73     | 1.35   |
| Interpolation        | 0.5    | 0.2      | 0.3    | 0.2    | 0.2    | 0.1      | 0.1    |
| Instability (sensor) | 0.04   | 0.03     | 0.02   | 0.01   | 0.01   | 0.02     | 0.01   |
| Back-reflection      | 0.1    | 0.1      | 0.1    | 0.1    | 0.1    | 0.1      | 0.1    |
| Alignment            | 0.1    | 0.1      | 0.1    | 0.1    | 0.1    | 0.1      | 0.1    |
| Nonlinearity         | 0.2    | 0.15     | 0.15   | 0.15   | 0.15   | 0.15     | 0.2    |
| Stray light (sensor) | 0.2    | 0.2      | 0.2    | 0.2    | 0.2    | 0.2      | 0.2    |
| Temperature          | 0.02   | 0.01     | 0.01   | 0.03   | 0.09   | 0.2      | 0.38   |
| Instability (source) | 0.14   | 0.14     | 0.12   | 0.11   | 0.1    | 0.09     | 0.08   |
| Uniformity           | 0.1    | 0.1      | 0.1    | 0.1    | 0.1    | 0.1      | 0.1    |
| Stray light (source) | 0.1    | 0.1      | 0.1    | 0.1    | 0.1    | 0.1      | 0.1    |
| Signal, type A       | 0.12   | 0.07     | 0.04   | 0.02   | 0.03   | 0.03     | 0.06   |
| Combined ($k = 1$)   | 0.64   | 0.41     | 0.46   | 0.39   | 0.40   | 0.40     | 0.53   |
| Expanded ($k=2$)     | 1.3    | 0.8      | 0.9    | 0.8    | 0.8    | 0.8      | 1.1    |

## 4.3. Uncertainty Components in Tables 3 and 4

### 4.3.1. Calibration Certificate

Calibration certificate of the radiometer provides calibration points of radiometric responsivity following the individual wavelength scale of the radiometer. This component is presented only for reference and is not included in the combined and expanded uncertainties.

### 4.3.2. Interpolation

Interpolation of radiometer's data is needed due to differences between individual wavelength scales of the radiometers. Therefore, measured values were transferred to a common scale basis (Sentinel-3/OLCI bands) for comparison, see 3.3. The uncertainty contribution associated with interpolation of spectra is estimated from calculations using different interpolation algorithms. The weights used for binning hyperspectral data to OLCI bands depend on the wavelength scale and exact pixel positions of the hyperspectral sensor. In Table 3, the interpolation components include the contribution of wavelength scale uncertainty estimated from data presented in Figure 14 of [3].

### 4.3.3. Temporal Instability of Radiometer

Instability of the radiometric responsivity can be estimated from data of repeated radiometric calibrations. For LCE-2, the instruments were calibrated just before the comparisons and only short-term instability relevant for the time needed for the measurements has to be considered. The values are derived from the data collected in calibration sessions of LCE-2 and FICE-AAOT a year later, see Section 4.1.1 and Figure 9. The variability over two weeks was interpolated from the yearly variability data. In addition to instability of the sensors the data in Figure 9 includes other uncertainty components related to the calibration setup (e.g., alignment, short-term lamp instability, etc.).

### 4.3.4. Back-Reflection

Back-reflection from the radiometer into the integrating sphere was estimated using different distances between the sphere and the radiometer as contribution of radiation reflecting from the radiance sensor back into the integrating sphere.

### 4.3.5. Polarization

The polarization effect was estimated by indoor irradiance measurements, repeating cast after radiometer was rotated 90° around its main optical axis, and revealing so the combined effect of alignment and polarization. According to [38] the FEL emission is polarized about 3%. As reported in [24], the polarization sensitivity of RAMSES irradiance sensors is varying from (0.05 . . . 0.3)% at 400 nm to (0.3 . . . 0.6)% at 750 nm. Due to the depolarizing nature of the cosine collector this effect is smaller than polarization sensitivity of RAMSES radiance sensors. Therefore, the observed differences with rotated sensors are mostly caused by other effects like alignment, instability of measured source, etc., and from the indoor irradiance uncertainty budget the polarization component is omitted. Polarization is also not included in the indoor radiance uncertainty budget as the integrating sphere is a strong depolarizer.

### 4.3.6. Alignment

Evaluation of alignment errors includes determination of distance between the source and the reference plane of the cosine collector, measured along the optical axis. Alignment includes also position errors of the lamp source across optical axes, rotation errors of the lamp [39], and positioning errors of the input optics of the radiometer. Combined alignment and positioning errors are included in variability data of radiometers calibrated with two different lamps (Figure 9).

### 4.3.7. Nonlinearity

Due to nonlinearity, some hyperspectral radiometers, measuring at different integration times, may show relative differences up to 4% (see Figure 12 in Section 4.1.4). According to recommendations, the non-linearity effects of good sensors should be correctable to less than 0.1%. The non-linearity correction (3) was applied to both calibration and measurement spectra, with residues expected to be less than 0.2%.

### 4.3.8. Spectral Stray Light

Spectral stray light of sensors is commonly not very relevant for measurements when the calibration and target source emissions have similar spectral composition. Value is estimated from Figure 14 in Section 4.1.5, and from [22,32].

### 4.3.9. Temperature

For array spectroradiometers with silicon detectors, the present estimate of standard uncertainty due to temperature variability (±1.5 °C) in the spectral region from 400 nm to 700 nm is around 0.1% and will increase up to 0.6% for longer wavelengths (950 nm) [13].

### 4.3.10. Temporal Instability of Radiation Source

The short-term instability of the source is relevant for the indoor measurements as they were not made simultaneously by all the participants. Thus, the time needed for intercomparison measurements, including power cycling the source between the two days of indoor experiment, has to be considered. This uncertainty component was estimated using the uncertainty in setting the lamp current and its effect on lamp emission. The drift of the irradiance values (at 500 nm) measured during the calibration campaign was ~0.1%, (2.2).

### 4.3.11. Stray Light in Laboratory

Sources of stray light are associated with the stray light in the laboratory during the indoor experiment. This component has been estimated in previous experiments made in the Tartu Observatory.

### 4.3.12. Type A Uncertainty of Repeated Measurements

For Type A uncertainty of time series of indoor measurements, white noise of measured series can be often expected. The analysis has indicated that sometimes the measurements are not completely independent and the autocorrelation of time series has been accounted for. If there is autocorrelation in the time series, the effective number of independent measurements $n_e$ has to be considered instead of actual number of points $n_t$ in the series [40]

$$n_e \approx n_t \frac{1 - r_1}{1 + r_1},\tag{7}$$

where $r_1$ is the lag-1 autocorrelation of the time series.

## 5. Discussion and Conclusions

The LCE-2 exercise consisted of three sub-tasks: SI-traceable radiometric calibration of participating radiometers just before the intercomparison; laboratory intercomparison of stable lamp sources in controlled environment; outdoor intercomparison of natural radiation sources over terrestrial water surface. Altogether, 44 radiometric sensors from 11 institutions were involved: 16 RAMSES, 2 OCR-3000, 4 HyperOCR, 4 WISP-3, 1 SeaPRISM and 1 SR-3500 radiance sensors, and 10 RAMSES, 1 OCR-3000, 2 HyperOCR, 2 WISP-3, and 1 S R-3500 irradiance sensors. Additionally, the majority of sensors involved in LCE-2 were recalibrated at TO a year later (for FICE-AAOT) giving estimate for their long-term stability. More than 80% of the sensors changed during one year less than ±1%.

Agreement between the radiometers is mostly affected by the calibration state of sensor. For example, factory calibrations made at different times can cause differences exceeding ±10%. Former calibrations in different laboratories from several years ago can cause differences around ±3%. Different calculation schemes (corrections for non-linearity, stray light or for OLCI band values) can cause differences about ±1 ... 2% each factor. The best agreement of 0.5 ... 0.8% between participants has been achieved when measurements were carried out just after calibration and for data handling unified procedures have been used including application of nonlinearity correction and the same algorithm for calculation of OLCI band values.

Dependence of the calibration coefficients on temperature can also cause significant deviation from SI-traceable result, especially in the near-infrared spectral region. For maximum temperature difference of about 20 °C between calibration and later measurements (typically between 0 °C and 40 °C) a responsivity change more than 10% is possible [2,13]. For laboratory measurements in controlled environment the temperature effect is expected to be within (0.1 ... 0.5)%.

Effect of stray light correction evaluated for indoor measurements in the range (400 ... 700) nm has been less than 0.5%. Though, outside the range of (400 ... 700) nm the relative uncertainty may increase substantially if correction is not applied.

Maximum value of the nonlinearity effect due to integration times determined from calibration spectra of TriOS RAMSES radiometers for a group of 15 radiometers was in the range of (1.5...4)%. At the same time, variability between the instruments due to this effect if not corrected, remained within ±1% due to the systematic nature of the nonlinear behavior affecting all the instruments in similar manner. During laboratory measurements the non-linearity correction was applied to both calibration and measurement spectra, with residues expected to be less than 0.2%.

Field of view and cosine responsivity effects can significantly depend on the limits of error set by specifications of radiometers, and on results of individual tests showing how large is deviation from the specified values. In the laboratory, the cosine responsivity error of the sensor during calibration was close to the error during the intercomparison measurements due to similar illumination geometry, and therefore, the resulting systematic error is insignificant.

Through the indoor experiment, when conditions for later measurements and conditions specified for calibration were quite similar, high effectiveness of the SI-traceable radiometric calibration has been demonstrated, a large group of different type radiometers operated by different scientists achieved satisfactory consistency between results showing low standard deviations between radiance (27 in total) or irradiance (15 in total) results (s < 1%). This is provided when some unification of measurement and data processing was settled: alignment of sensors, structure of collected data, application of unified wavelength bands and non-linearity corrections. Nevertheless, variability between sensors may be insufficient for complete quantification of uncertainties in measurement. For example, standard deviation of nonlinearity estimates versus the mean effect (Figure 13) demonstrates that differences are not able to reveal full size of systematic errors common for all the instruments. Therefore, all radiometers still should be individually tested for all significant systematic effects which may affect the results as this is the only way to get full estimate of the effects degrading traceability to the SI scale.

Furthermore, in the frame of the outdoor experiment when conditions specified for calibration and in field are much more different from each other the variability between freshly calibrated individual sensors did increase substantially, demonstrating the limitation of typical OC field measurements with sensors having SI-traceable radiometric calibration. Including laboratory intercomparison to the LCE-2 exercise has clearly shown that further reduction of the uncertainty of radiometric calibration of sensors will not improve the agreement between field results significantly. Much more relevant for achieving better SI-traceability of field measurements are improved specifications of radiometers, additional characterization of individual sensors accounting for specific field conditions, and unified data handling which will be considered in [1].

**Author Contributions:** Conceptualization, V.V., J.K., I.A., R.V., K.A., K.R., C.D., T.C.; Data curation, V.V., I.A.; Formal analysis, V.V., J.K., I.A.; Investigation, J.K., I.A., R.V., K.A., K.R., A.A., M.B., H.B., M.C., D.D., G.D., B.D., T.D., C.G., K.K., M.L., B.P., G.T., R.D., S.W.; Methodology, V.V., J.K., I.A., K.A., K.R.; Project administration, R.V.; Software, I.A.; Validation, V.V., J.K., I.A.; Visualization, V.V., J.K., I.A., K.R., M.L.; Writing—original draft, V.V., J.K., I.A., R.V., K.A.; Writing—review & editing, V.V., J.K., I.A., R.V., K.A., K.R., A.A., M.B., H.B., M.C., D.D., G.D., B.D., T.D., C.G., K.K., M.L., B.P., G.T., R.D., S.W; A.B., C.D., T.C.

**Funding:** This work was funded by European Space Agency project Fiducial Reference Measurements for Satellite Ocean Color (FRM4SOC), contract no. 4000117454/16/I-Sbo.

**Acknowledgments:** Support from numerous scientists, experts, and administrative personnel contributing to the project are gratefully acknowledged. The authors express special gratitude to Giuseppe Zibordi (JRC of the EC), Anu Reinart (UT), Tiia Lillemaa (UT), Mari Allik (UT), Claire Greenwell (NPL).

**Conflicts of Interest:** The authors declare no conflict of interest. Most authors collecting experimental data are customers of the respective instrument manufacturers. Two authors (R.V.D. and B.D.) are employees of radiometer manufacturers. Two authors (J.K. and K.R.) are developing a new radiometer for commercialization in 2021. Study data analysis and conclusions focus on aspects which are common to all radiometers and results are anonymized to avoid any risk of bias. This study does not constitute endorsement of any of the products tested by any of the authors or their organizations.

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
