# Peer review of "Laboratory Intercomparison of Radiometers Used for Satellite Validation in the 400–900 nm Range"

_remotesensing, doi:10.3390/rs11091101_

Round 1
Reviewer 1 Report
General comments
The work reports an inter-comparison exercise for ocean color radiometers operating between 400-900 nm. The motivation of the work is to assess the consistency among measurements performed by different instruments and people while exploring different sources of uncertainty, comparing field protocols and data treatment. This particular work actually focused on instrument performances and their stability under laboratory conditions, where conditions differ largely from what is found in the field. This implies on the need for a number of data corrections, and little have been discussed about that so far in the literature.
In this 2-step exercise, involving 11 institutions 44 instruments and 5 manufacturers, techniques and procedures evaluated the retrieval of data simulating the OLCI bands. During step 1, all sensors were calibrate in the same calibration facility, and during step 2, the conditions were replicated, but this time, calibrations were conducted by the authors. Comparisons among indoor conditions included thermal effects, nonlinearity, stray light, angular response, and polarization. Results showed mainly the importance of calibration state of the sensors and the data handling, and suggest that given proper procedures, the differences among instruments are not an issue for future algorithms, at least for lab conditions.
The work is indeed important and interesting, and it is very well written. However, a little more work is still necessary for the discussion section, the weakest part of the work, which rather summarized the results instead of discussing them. I understand other publication [1] is being prepared for the same special issue showing the results of the third step of the exercise, but the links between the two papers are missing here and should be better addressed.
Minor comments:
Line 63 - Replace LCE-2 by The 2nd 24 Laboratory Comparison Exercise (LCE-2)
Lines 108-114: I do not see the need for bullets.
Line 189 - was a threshold value used for the data of the two-channel optical sensor? Are these the same as the ones described in lines 235-238?
Line 212- Were the diffusers cleaned before the measurements?
Line 309- I am not sure about the source of the discrepancies, how were them identified?
Line 415 - This is an important point of the paper. A flow diagram may be helpful.
Suggestions:
Title- It would be helpful having the words "correction” and “data handling” there
Figures - Please prepare new figures. At least in my pdf version they look very rough and unprofessional. Try avoiding dark backgrounds as in Figure 1, 3 and 4. I liked the photographs of all sensors lined up, but not those in Figure 1. Perhaps they could be supplementary materials?.
The other Figures on the results section can also be improved by presenting less lines per figure, perhaps by diving the instruments into groups in different sub graphs rather than using different line types in a single one.
References (I could not find the text below).
[1] V. Vabson et al., ‚Field intercomparison of radiometers used for satellite validation in the 400 - 900 nm range, Remote Sens., vol. Special Issue ‚Fiducial Reference Measurements for Satellite Ocean Colour, Mar. 739 2019.
Author Response
Response to Reviewer 1 Comments
General comments
The work reports an inter-comparison exercise for ocean color radiometers operating between 400-900 nm. The motivation of the work is to assess the consistency among measurements performed by different instruments and people while exploring different sources of uncertainty, comparing field protocols and data treatment. This particular work actually focused on instrument performances and their stability under laboratory conditions, where conditions differ largely from what is found in the field. This implies on the need for a number of data corrections, and little have been discussed about that so far in the literature.
In this 2-step exercise, involving 11 institutions 44 instruments and 5 manufacturers, techniques and procedures evaluated the retrieval of data simulating the OLCI bands. During step 1, all sensors were calibrate in the same calibration facility, and during step 2, the conditions were replicated, but this time, calibrations were conducted by the authors. Comparisons among indoor conditions included thermal effects, nonlinearity, stray light, angular response, and polarization. Results showed mainly the importance of calibration state of the sensors and the data handling, and suggest that given proper procedures, the differences among instruments are not an issue for future algorithms, at least for lab conditions.
The work is indeed important and interesting, and it is very well written. However, a little more work is still necessary for the discussion section, the weakest part of the work, which rather summarized the results instead of discussing them. I understand other publication [1] is being prepared for the same special issue showing the results of the third step of the exercise, but the links between the two papers are missing here and should be better addressed.
Response 1: Main content of the article is precisely summarized. We thank the reviewer for helpful remarks and comments.
Minor comments:
Line 63 - Replace LCE-2 by The 2nd 24 Laboratory Comparison Exercise (LCE-2)
Response 2: Accepted.
Lines 108-114: I do not see the need for bullets.
Response 3: Accepted.
Line 189 - was a threshold value used for the data of the two-channel optical sensor? Are these the same as the ones described in lines 235-238?
Response 4: The sensors are different. The two-channel sensors are used for the simultaneous monitoring of the stability of lamps during their use. Filter radiometer based on the silicon trap detector is used to reveal the changes of the lamps before and after the comparison exercise.
Line 212- Were the diffusers cleaned before the measurements?
Response 5: The diffusers have been cleaned before the calibration.
Line 309- I am not sure about the source of the discrepancies, how were them identified?
Response 6: If significant difference was present, we repeated the calculations at TO applying the same algorithm to the raw measurement data of the discrepant participant, and by using the latest calibration data file. In this way the likely reason of discrepancy, e.g. the use of wrong calibration data, calculation algorithm of OLCI band values, application nonlinearity corrections etc., was revealed.
Line 415 - This is an important point of the paper. A flow diagram may be helpful.
Response 7: We agree with the comment however, we also believe that the diagram would be too complicated and confusing within the context of the present paper. A paper dedicated on uncertainty evaluation is planned within the current MDPI special issue.
Suggestions:
Title- It would be helpful having the words "correction” and “data handling” there
Response 8: Could we ask you to be more specific.
Figures - Please prepare new figures. At least in my pdf version they look very rough and unprofessional. Try avoiding dark backgrounds as in Figure 1, 3 and 4. I liked the photographs of all sensors lined up, but not those in Figure 1. Perhaps they could be supplementary materials?.
Response 9: The resolution of Figures 1, 3 and 4 has been enhanced, the backgrounds of figures modified.
The other Figures on the results section can also be improved by presenting less lines per figure, perhaps by diving the instruments into groups in different sub graphs rather than using different line types in a single one.
Response 10: We agree that specific behaviour of individual sensors will be of certain interest, especially for their owners/users. For other readers general effects hopefully are more interesting: qualitative improvement of agreement between sensors as a result of radiometric calibration and unified data handling is clearly demonstrated.
Reviewer 2 Report
This manuscript describes the activities and results of the SI-traceable radiometric calibration and indoor intercomparison of radiance and irradiance ocean colour radiometers.Reasons that can influence the consistency of measurements from different instruments are analysed. The discussion and results are important and can be interesting to the related scientific community. The manuscript is acceptable for publication.
Author Response
Response to Reviewer 2 Comments
Comments and Suggestions for Authors
This manuscript describes the activities and results of the SI-traceable radiometric calibration and indoor intercomparison of radiance and irradiance ocean colour radiometers.Reasons that can influence the consistency of measurements from different instruments are analysed. The discussion and results are important and can be interesting to the related scientific community. The manuscript is acceptable for publication..
Response 1: The content of the article is well described. We thank the reviewer for helpful comments.
Reviewer 3 Report
This manuscript covers the laboratory intercomparison of radiometers used for validation of satellite imagery. As such, it's an important contribution. The intercomparison of instruments used for calibration and validation efforts is a key link in the remote sensing processing chain to ensure that we are retrieving validated physical measurements.
That said, I think the authors could make the abstract shorter and the introduction is a little disorganized. The second introductory paragraph makes a better start than the first, but is also far too long to be a single thought.
In some places there are issues with English - for example, L100 - "as regards of" should be "in regards to the..." The paragraph that starts on page 4 is also multiple paragraphs. It does hamper the readability of the paper.
L300 - ASCII is a means of holding text files and is quite distinct from an MS Access database. So these things are not synonyms. You mean to say the data were stored as a text file or in an Access database. It's clear what was done, but just a little too much detail on the wrong thing. It's important to know what was stored and how it was organized and used. You can't really define a file format as a use for calculations.
The rest of the paper is excellent.
Author Response
Response to Reviewer 3 Comments
Comments and Suggestions for Authors
This manuscript covers the laboratory intercomparison of radiometers used for validation of satellite imagery. As such, it's an important contribution. The intercomparison of instruments used for calibration and validation efforts is a key link in the remote sensing processing chain to ensure that we are retrieving validated physical measurements.
Response 1: We agree with Reviewer.
That said, I think the authors could make the abstract shorter and the introduction is a little disorganized. The second introductory paragraph makes a better start than the first, but is also far too long to be a single thought.
Response 2: We made the abstract and introduction slightly shorter. The 2nd Laboratory Comparison Exercise - LCE 2 – is a small piece in a big picture of Fiducial Reference Measurements. In introductory paragraph we tried to give an overview about basic and general concepts relevant to the described intercomparison exercise, surely less interesting for specialists well informed about all these issues.
In some places there are issues with English - for example, L100 - "as regards of" should be "in regards to the..." The paragraph that starts on page 4 is also multiple paragraphs. It does hamper the readability of the paper.
Response 3: Accepted.
L300 - ASCII is a means of holding text files and is quite distinct from an MS Access database. So these things are not synonyms. You mean to say the data were stored as a text file or in an Access database. It's clear what was done, but just a little too much detail on the wrong thing. It's important to know what was stored and how it was organized and used. You can't really define a file format as a use for calculations.
Response 4: Accepted. Device-specific technical details show how the general use of unified protocol still may result in specific differences between the instruments.
The rest of the paper is excellent.
Response 5: Thank you!
Reviewer 4 Report
LINE 210: TO DESCRIBE IN DETAIL THE DISTANCE MEASUREMENT METHOD
LINE 213: IT IS NECESSARY TO SPECIFY THE MEASUREMENT ERROR
LINE 223: IT IS NECESSARY TO INDICATE THE ACCURACY OF MEASURING VARIATIONS OF TEMPERATURE, HUMIDITY AND ATMOSPHERIC PRESSURE
IN EACH SPECIFIC CASE, IT IS NECESSARY TO INDICATE A REASONABLE NUMBER OF THE TAKEN MEASUREMENTS. FOR EXAMPLE, LINE 249: WHY TWICE, NOT THREE TIMES, ETC. THE ACCURACY OF MEASUREMENTS IMPROVES WITH AN INCREASE IN THE AMOUNT OF MEASUREMENTS.
Author Response
Response to Reviewer 4 Comments
Comments and Suggestions for Authors
LINE 210: TO DESCRIBE IN DETAIL THE DISTANCE MEASUREMENT METHOD
Response 1: The method is described in details (chapter 2.2, lines 207..218) and Figure 3. The distance probe is calibrated by using a SI-traceable 500 mm micrometer standard.
LINE 213: IT IS NECESSARY TO SPECIFY THE MEASUREMENT ERROR
Response 2: The stability of filter radiometer is largely determined with the instability of bandpass filters, which did change during the two week period needed for calibrations less than 0.1 %. The type A component of trap detector is less than 0.02 % in the range of 400…900 nm. The contribution due to long term instability of the trap detector is insignificant.
LINE 223: IT IS NECESSARY TO INDICATE THE ACCURACY OF MEASURING VARIATIONS OF TEMPERATURE, HUMIDITY AND ATMOSPHERIC PRESSURE
Response 3: The temperature and humidity sensors are calibrated at TO by using SI-traceable standards. The expanded uncertainty (k=2) of sensors used for the monitoring of environmental conditions is the following: 0.7 °C for temperature sensors, and 5 % RH for humidity sensors. The pressure sensor is used as indication only.
IN EACH SPECIFIC CASE, IT IS NECESSARY TO INDICATE A REASONABLE NUMBER OF THE TAKEN MEASUREMENTS. FOR EXAMPLE, LINE 249: WHY TWICE, NOT THREE TIMES, ETC. THE ACCURACY OF MEASUREMENTS IMPROVES WITH AN INCREASE IN THE AMOUNT OF MEASUREMENTS.
Response 4: As for this project NPL provided two standard lamps, full re-alignment of the calibration setup was necessary for each calibrated radiometer. Thus, each radiometer was calibrated twice against two different lamps. The calibration process performed by using each lamp was identical, containing at least 30 repeated acquisitions, except for WISP-3 and CIMEL, in which case the high number of acquisitions was technically unreasonable.
Switching on and off the standard lamp source is extremely costly and during the radiometric calibration of such instruments normally reduced to minimum. Therefore, quite often the alignment uncertainty is assessed by using similar non-calibrated sources.
Round 2
Reviewer 4 Report
Article has average scientific level, but it can be published.